# Bone Modelling and Remodelling in Cold Environment

**DOI:** 10.3390/biom15040564

**Published:** 2025-04-11

**Authors:** Leyi Xue, Qiao Guan, Lingli Zhang

**Affiliations:** 1College of Athletic Performance, Shanghai University of Sport, Shanghai 200438, China; 22210614@sus.edu.cn; 2School of Exercise and Health, Shanghai University of Sport, Shanghai 200438, China; 2321518016@sus.edu.cn

**Keywords:** cold environment, bone, bone microenvironment, bone vascularity, bone fat

## Abstract

People engaged in various activities in cold environments—such as those living in cold climates, polar workers, cold storage workers, and athletes engaged in winter sports—are frequently affected by cold environments. Therefore, it is of great significance to explore the modelling and remodelling of bones in cold environments. Cold environments can shorten the length of bones, thin the thickness of bones, decrease bone mineral density (BMD), change the biomechanical properties of bones, and lead to bone loss. In addition, cold directly affects the bone microenvironment. Exposure to cold causes spindle-like and fibroblast-like changes in bone marrow mesenchymal stem cells (BMSCs) and decreases their proliferation, and cold exposure promotes the osteogenic differentiation of BMSCs partly through the *p38 MAPK* pathway. Cold also alters the dendritic differentiation of OBs by reducing the transmembrane glycoprotein E11/podoplanin and damages endothelial cells (ECs) by elevating levels of VEGF, resulting in a reduced blood supply and thus fewer OBs. In addition, cold promotes lipolysis of marrow adipose tissue (MAT), but in combination with exercise, it can promote the differentiation of BMSCs into MAT. Cold environments interfere with angiogenesis and inhibit bone growth by affecting factors such as platelet-derived growth factor type BB (PDGF-BB), slit guidance ligand 3 (SLIT3), *Notch*, and VEGF. In addition, cold environments may promote bone resorption by activating sympathetic nerves to activate β-adrenergic receptors and regulating leptin secretion, and regulate bone metabolism by activating the *p38 MAPK* signalling pathway and increasing the synthesis of brown fat, which ultimately inhibit bone formation and enhance bone resorption. In this paper, we describe the effects of cold environments on bones in the locomotor system in terms of bone structure, bone mass, biomechanical properties, and various skeletal cells, bone blood vessels, and bone fat systems in the bone microenvironment.

## 1. Introduction

Cold environments are defined as conditions that cause the body’s skin temperature or body temperature to drop, causing the body to respond with shivering, inhibition of sweating, and vasoconstriction of skin blood vessels to maintain core temperature [1]. For example, a cold snap is a type of cold environment; it is a specific type of extreme atmospheric event that manifests itself as several consecutive days of abnormally low temperatures [2]. In the United States, cold weather such as this has been identified as the leading cause of weather-related deaths, with more than 60% of weather-related deaths estimated to be attributable to cold weather during the period from 2006 to 2010 [3]. This shows that cold environments have a significant impact on human health.

Bone is one of the most important support structures in the body, providing extensive attachment points for muscles and influencing systemic homeostasis through the release of endocrine factors and calcium. Its health directly affects an individual’s physical function and longevity [4]. Bone has a developmental plasticity response to temperature, and prolonged exposure to cold temperatures can damage skeletal structure [5]. Pienimaki [6], through a questionnaire survey, found that up to 90% of respondents in Finland (average year-round temperature of 0–7 °C) experienced cold-related musculoskeletal pain (CMP) at −23 °C, with 30% of these respondents experiencing CMP in at least one body part. Similarly, Farbu [7] used data from the sixth survey of the Tromsø Study, analysing 6533 subjects, and found that cold environments triggered pain in ≥3 sites. Respondents who worked in the cold experienced pain in all sites except the hands, feet, stomach, and head. For example, Zeng [8] found a dose-response relationship between working in the cold and the risk of developing rheumatoid arthritis. This suggests that bones are affected in people who work or live in cold climates and environments for long periods of time.

Through a review of the literature, we summarised the effects of cold environments on bone phenotype, clarified the limiting effect of cold on bone growth and development, and elaborated on the changes in the bone microenvironment from the perspective of molecular mechanisms such as *p38 MAPK* signalling pathway, RANKL, *Runx2*, and others. The study of the functional changes in the skeleton due to cold environments is of great significance for the prevention of osteoporosis in people living in cold climates or in specialised occupations (e.g., polar workers, cold storage workers). Additionally, it can improve the competitive level of athletes in ice sports and help prevent sports injuries.

## 2. Effect of Cold Environment on Bone Mass

Cold environments affect bone length, which is closely related to height. Height is a polygenic trait [9], which is determined by the longitudinal growth of limb bones and vertebrae [10]. The bones of the limbs and vertebrae grow longitudinally. There are approximately 206 bones in the body, and most of the lower limb bones, such as the femur, tibia, and fibula, play a major role in height growth, so bone length plays an important role in height [11]. Allen [12] first proposed Allen’s Law, which states that animals living in colder regions tend to have shorter limbs than their warmer counterparts. This was further demonstrated by Doucette [4], who activated the chronic sympathetic nervous system through cold exposure at 4 °C, which showed shorter bones in mice fed at low temperature (4 °C) compared to the controls at room temperature (22 °C). The study results of Al-Hilli [13] tend to be consistent, showing that the tail bones of mice fed at a higher ambient temperature (33 °C) are longer than those fed at a lower ambient temperature (8 °C). It was also found that the free tail vertebrae of mice kept at 33 °C were longer and rounder than those kept at 8 °C, while the middle part of the backbone of the mice kept at 8 °C was shorter and thinner [14]. In addition, Robbins [5] found that cold chamber (20 °C) mice had shorter bodies and tails, lower trabecular volume fractions, thinner distal femoral trabeculae, and fewer connections; they also showed a tendency for lower and middle femur cortical bone volume fractions, and a trend towards lower cortical thickness, as compared to mice kept close to thermoneutrality (26 °C). In conclusion, parameters such as skeletal cortical bone and bone trabeculae were negatively correlated with hypothermia [5]. Therefore, bones are generally shorter in cold environments, and the mechanism for analysing this phenomenon is firstly explained by the vascularity hypothesis [15]. Cold causes vasoconstriction of blood vessels as a form of insulation [16]. However, some studies do not fully support the vascular hypothesis, with Serrat [17] suggesting that blood not only provides essential growth factors, nutrients, and oxygen to the developing limb but also serves as an important heat source [18] and that chondrocyte proliferation and matrix production are affected by blood temperature. Vasoconstriction decreases the temperature of the developing cartilage, chondrocyte proliferation is reduced, matrix is destroyed, and the rate of skeletal growth is slowed according to the model of limb bone as an endochondral osteogenic formation. Consistent with this, Bierens de Haan [19] found that low temperatures have a direct lethal effect on chondrocytes in the epiphyseal growth plate, and that brief exposure to −20 °C to −15 °C resulted in premature growth arrest of the epiphyseal plate, epiphyseal disruption, and reactive periosteal and osteochondral bone formation, all of which led to significant shortening and widening of the skeleton. At the same time, Serrat [20] also found that solute transport to the avascular cartilage plate is also critical for bone growth. Longitudinal bone growth occurs in the cartilaginous plate, which, unlike other organs, requires nutrient support through solute transport from the vascular system, a process directly influenced by temperature. Serrat [20] found that solute delivery in warm (36 °C) limbs was twice as high as that in cold limbs (23 °C), confirming that a cold environment would inhibit solute delivery, thereby limiting cartilage plate’s access to nutrients and thus slowing longitudinal bone growth. In addition, Serrat [20] also found that exercise attenuated the effects of hypothermia on limb growth retardation, with all running mice having significantly longer limbs and non-running mice having shorter limbs, regardless of temperature.

Cold environments can also affect the bone surface area. The higher the latitude of a country, the further it is from the equator, and the colder it is [21]. Roberts [22] found that compared with low-latitude people, high-latitude people tend to have wider and larger bodies and shorter distal limb segments. According to Bergman’s law and Allen’s law, latitude is positively correlated with body weight and bi-iliac width, and negatively correlated with surface area/mass ratio and distal limb segment lengths [23]. Robbins [5] also confirmed that, compared with mice kept at 20 °C, mice at 26 °C had a significantly reduced distal femur trabecular bone volume fraction, trabecular bone density, thickness, and connectivity density (Conn.D) The femoral cortical bone in the middle area was also lower. Moreover, exposure to cold temperatures during rapid bone growth increased sympathetic tone, resulting in trabecular microstructural damage, altered cortical bone cross-sectional geometry, and weakened bone strength [5]. Consistent with this, Liu [24] also found that bone volume fraction, trabecular number, trabecular thickness, and connectivity density were reduced, trabecular separation was increased, and cross-sectional thickness was reduced in arthritic rats in a cold environment (5 ± 2 °C).

In addition, cold environments reduce BMD. People living in cold regions generally have lower BMD [25]. Liu [24] found that exposure to low temperatures (5 ± 2 °C) led to a decrease in BMD. Thompson [26] also found that adult *Eskimos* from colder regions (St. Lawrence Island, AK, USA; Yukon Territory, Canada; Nunavut, Canada) had faster bone mineral loss and lower BMD compared to *white Americans*. Reduced BMD in cold environments is associated with multiple mechanisms. First, cold exposure affects the production and release of neurotransmitters, such as neuropeptide Y (NPY), which inhibits energy expenditure, Uncoupling Protein-1 (UCP1) levels, and reduces bone mass under cold exposure conditions [27]. In addition, cold activates the sympathetic nervous system, increases sympathetic tone, and leads to bone loss through hypothalamic signalling of β2-adrenergic receptors on OBs [28]. Interestingly, brown adipose tissue (BAT) may also mitigate bone loss by participating in the regulation of sympathetic tone. When the body is in a cold environment, UCP1 activation in BAT shifts the oxidation of fatty acids and glucose to generate heat, reducing sympathetic tone and alleviating bone loss. This regulation helps to protect BMD, but is still insufficient to combat the negative effects of cold exposure on cortical bone [29]. At the same time, the body regulates adaptive thermogenesis and energy homeostasis in order to adapt to the cold environment. A study in humans also reported a positive correlation between BAT and BMD [30]. BAT non-shivering thermogenesis increased proportionally with cold exposure, but it was not sufficient to prevent bone loss [5]. However, Sanchez-Delgado [31] concluded that there is no association between cold-induced BAT and BMD, and that the previously shown association between BAT and BMD may be due to methodological issues related to BAT assessment and/or sample size limitations.

Under normal temperatures, bone mass maintenance relies on mechanical loading and endocrine regulation. PTH and vitamin D levels remain stable, supporting calcium-phosphate metabolism and bone formation. Cold exposure disrupts this equilibrium [32].

In summary, cold environments cause bone length to become shorter, bone thickness to become thinner, and BMD to decrease, leading to bone loss (Figure 1).

## 3. Effect of Cold Environment on Bone Biomechanics

The key biomechanical properties of bone include stiffness, toughness, ductility, and mechanical strength, which are referred to as the intrinsic biomechanical properties of bone when measured in bone tissue; whereas the extrinsic biomechanical properties reflect the structural behaviour of the entire bone. A good balance between the various intrinsic and extrinsic biomechanical properties of bone is essential for fracture resistance. Brazda [33] found that 16 cases of transverse cracking, three cases of oblique cracking, and one case of surface depression were observed in 20 cases of frozen porcine ribs, compared to thawed porcine ribs. Consistent with this, Pelker [34] found that freeze-drying (−20 °C, −70 °C, and −196 °C) caused severe damage to the torsional strength of long bones, with a significant reduction in torque, energy absorbed, and torsional stiffness at destruction, as well as a significant increase in the angle of destruction. Lee [35] similarly found that five years of freezing (−20 °C) reduced the ultimate strength of trabecular bone.

The main components of bone are hydroxyapatite, collagen, and water [36]. The modulus of elasticity of hydroxyapatite, which makes up 60–70% of the bone dry weight composition, is constant at 63 GP over a wide temperature range (−58 °C–−25 °C). The modulus of elasticity of collagen fibres, which make up 25–30%, rises from 1.2 GPa (25 °C) to 11 GPa (−18 °C). The modulus of elasticity of water, which makes up 25%, rises from 0 GPa (23 °C) to 9.3 GPa (−16 °C) [37], and when water changes from a liquid to a solid frozen state, it increases in volume by 9% [38]. Individual components of the whole bone respond differently to temperature, and a cold environment improves the elasticity and increases the volume of some of the bone constituents, providing mechanical support to the surrounding tissues. Secondly, the inherent hierarchical structure of the bone extracellular matrix (ECM) has specific microstructural features and energy dissipation mechanisms at different length scales, enabling bone to effectively resist the applied loads of different components [39]. The mineralised ECM has unique biomechanical properties to support body weight and exercise and serves as a source of essential minerals required for key bodily functions, with inorganic phosphate (Pi) playing an important role in regulating the mineralisation of the extracellular matrix. Dardenne [40] found that a reduction in vitamin D impairs intestinal absorption of Ca^2+^ and Pi. It is well known that the main source of vitamin D is the skin synthesis of cholecalciferol, which is dependent on UV radiation, and that cold fails to provide a significant amount of UV radiation, often leading to vitamin D deficiency [41]. Thus, being in a cold region may lead to vitamin D deficiency, which reduces the uptake of Pi, and this reduction in Pi leads to bone softening and ultimately affects the biomechanical properties of bone. Again, porosity has the most dominant effect on bone strength. Cortical porosity is negatively correlated with Young’s modulus [42,43]. Lee [35] found that the Young’s modulus and ultimate strength of bones frozen at −20 °C for 5 years were lower than those of fresh bone, which suggests that the cold environment decreases the Young’s modulus of bone and thus leads to an increase in porosity. In addition, based on the conclusions in the previous article, we know that cold causes bone loss, decreases BMD, and thus increases porosity, which ultimately affects the elastic modulus and strength of bone.

At normal temperatures, bones exhibit stable biomechanical performance with organised mineral content and collagen alignment, providing optimal mechanical support [44]. In contrast, cold conditions induce mineral loss and microstructural damage, significantly compromising mechanical strength.

For bone biomechanics, a cold environment reduces the intrinsic biomechanical properties of bone, such as strength and stiffness. In addition, the cold environment increases the elasticity and volume of some of the bone constituents, providing mechanical support to the surrounding tissues (Figure 2).

## 4. Effect of Cold Environment on Bone Metabolism

Bone is a highly dynamic tissue and organ that undergoes bone formation and bone resorption throughout its life. This process of successive replacement is known as bone metabolism, which typically involves three successive phases: the initiation of bone resorption by OCs; the transition from catabolism, dominated by OCs, to anabolism, dominated by OBs; and finally, the formation of the bone matrix by OBs and the replacement of old bone with new bone [45].

Du [46] found a significant decrease in bone formation rate/bone surface and an increase in the number of OCs/bone surface after 14 days of cold stimulation (4 °C). Similarly, Motyl [47] found that mice exposed to cold (4 °C) had a significant decrease in the percentage of mineralised surface, bone formation rate, number of OBs, and bone-like thickness, in addition to a significant increase in OC surface area, number of OCs and the percentage of eroded surface, suggesting that cold inhibits bone formation and promotes bone resorption.

Bone metabolism markers are particularly sensitive to cryogenic environments and change earlier than bone mass.

Straburzyńska-Lupa [48] studied the effects of single or multiple exposures to whole-body cryotherapy (WBC) at −110 °C on biomarkers of bone remodelling and blood-bone immune crosstalk in 22 healthy men. They found that sclerostin (SOST), a protein that downregulates bone formation and promotes bone resorption, soluble receptor activator for nuclear factor κ B ligand (sRANKL), which induces OCs formation and stimulates bone resorption, and the content of carboxy-terminal cross-linked telopeptide of type 1 collagen (CTX-1), which reflects the bone absorption activity of OCs, were all increased. They also observed an increase in osteocalcin (OCN), which reflects the activity of OBs. This is in agreement with Gombos [49], who suggested that bone resorption started immediately after stimulation, and that bone formation also started, albeit in a slightly shorter period of time. In addition, WBC can affect osteoclastogenesis inhibitory factor (OPG), an inhibitor of bone resorption. With WBC, OPG/RANKL increased in the high-fitness group and decreased in the low-fitness group. A single session of WBC can change the level of RANKL in the blood of a subject. However, only 10 or more sessions of whole-body cryotherapy resulted in differences in OCN, CTX-1, and OCN/CTX-1 between subjects [48]. In addition, Mu [50] found that parathormone (PTH) increased significantly after ice swimming (water temperature below 5 °C). Bone formation markers of procollagen type 1 N-terminal propeptide (P1NP) and C-terminal telopeptide of type 1 collagen (β-CTX) decreased significantly, suggesting that ice swimming inhibits bone synthesis and catabolism. Cold exposure promotes bone loss by inhibiting osteogenesis and enhancing osteoclastic metabolism [51]. It has been shown that sympathetic nerve activation by low temperature leads to β-adrenergic receptor activation and subsequent initiation of OCs-associated bone resorption [28]. However, the findings of cold exposure on bone formation and bone resorption are not entirely consistent, perhaps related to the temperature and duration of the cold exposure studied in the trials. Nie [52] showed that exposure to a cold environment at 4 °C for 7, 14, and 21 days could significantly up-regulate *Runx2* in the process of osteogenic differentiation of MSC, along with increased expression of Osteopontin (OPN), Bone sialoprotein (BSP) and Collagen I (Col1). At the same time, phosphorylation of the *p38 MAPK* pathway is activated to promote osteogenic differentiation of BMSCs.

Galliera [53] also found that elite football players were able to express OPN, BSP, and Col1 during a period of training in BMSCs, while activating the phosphorylation of the *p38 MAPK* pathway to promote BMSC osteogenic differentiation. Additionally, they found an increase in serum OPG/RANKL ratios in elite football players after a 5-day WBC (exposure to very cold air at −60 °C for 30 s in a temperature-controlled room with minimal clothing, followed by exposure to very cold air at −110 °C for 2 min). This complex phenomenon is explained by the fact that cold exposure induces changes in bone metabolism and morphology by increasing sympathetic tone and increasing BAT [5]. The regulation of bone metabolism by BAT is multifaceted. The cold environment stimulates BAT, which secretes cytokines such as fibroblast growth factor 21 (FGF21), which promotes bone resorption and inhibits bone formation, as well as interleukin 21, which accelerates bone resorption [54]. Similarly, interleukin-6 (IL-6) accelerates bone resorption [55]. However, cold stimulation promotes BAT to increase UCP1 expression to reduce sympathetic nerve levels and inhibit bone resorption [56]. Du [46] found that BAT upregulated alkaline phosphatase (ALP), Ocn, and *Runx2* levels, promoted osteogenic differentiation, and reduced OCs and their resorption capacity. In addition, BAT secretes leptin to regulate bone metabolism. Similar to BAT, the effect of leptin on bone metabolism is divided into two aspects. One is the peripheral effect of indirectly inhibiting bone resorption by regulating the function of OBs and OCs to ultimately cause an increase in bone mass and inhibiting inflammatory cytokines. The second is the central effect of inhibiting bone formation and enhancing bone resorption by uncoupling the bone remodelling unit from bone remodelling through the hypothalamus–sympathetic nerves to reduce bone mass [28]. Most studies have shown that the regulation of bone metabolism by leptin is negative [57]. In conclusion, the regulation of bone metabolism by BAT is complex, as it can accelerate bone resorption through the secretion of FGF21, IL-6, and leptin. On the other hand, the regulation of bone metabolism by BAT is positive in humans, upregulating the levels of ALP, OCN, and *Runx2*, and promoting osteogenic differentiation, and it can also reduce the resorption capacity of OCs and diminish their resorption capacity. In conclusion, BAT plays a very important role in the regulation of bone metabolism.

Normal-temperature bone homeostasis is maintained through coordinated activities of OBs and OCs. Bone formation markers (e.g., OCN, Col1) are highly expressed, while resorption markers (e.g., CTX-1) remain low [58]. Cold exposure shifts this balance toward enhanced OC activity.

Most experiments have shown that cold environments affect the expression of bone metabolism markers, such as an increase in the levels of SOST, which promotes bone resorption, sRANKL, which induces the production of OCs, and CTX-1, which reflects the activity of OCs. There is also a small increase in the concentration of OCN, which reflects the activity of OBs. In addition, the cold environment activates sympathetic nerves and β-adrenergic receptors to promote bone resorption and inhibit bone formation. A few experiments have also confirmed that short-term cold exposure has a positive effect on the regulation of bone metabolism. Cold exposure inhibits bone resorption by activating the *p38 MAPK* signalling pathway and increasing BAT synthesis. The role of BAT is complex; it can either secrete cytokines, such as leptin, FGF21, and IL-6, to promote bone resorption or promote bone formation and inhibit bone resorption by increasing the expression level of UCP1, as well as upregulating the levels of ALP, Ocn, and *Runx2* (Figure 3).

## 5. Effect of Cold Environment on Bone Microenvironment

### 5.1. Effect of Cold Environment on Skeletal Cells

The bone microenvironment serves as a support for the integration of the two major systems of bone and bone marrow, and mainly includes cellular components such as haematopoietic stem cells (HSCs), MSCs, OBs, bone lining cells, bone marrow adipocytes, bone macrophages, and immune cells, as well as the bone matrix, vascular structures, and other components [59]. The following is a list of the cellular components of MSCs.

BMSCs are adult stem cells with extensive self-renewal and clonal expansion capabilities, as well as the ability to differentiate into various tissue types and regulate the immune system [60]. BMSCs are a pluripotent cell type. Moreover, BMSCs are a pluripotent cell type that can give rise not only to OB lineage cells but also to a range of other cell types, including adipocytes [61]. The *mitogen-activated protein kinase (MAPK)* pathway has been reported to be involved in cell proliferation and differentiation and has been associated with osteogenic differentiation of BMSCs [62]. Nie [52] found that BMSCs from mice exposed to cold (4 °C) for 7, 14, and 21 days had spindle and fibroblast-like morphology, and the phosphorylation level of *p38 MAPK* was significantly upregulated, suggesting that cold exposure promotes osteogenic differentiation of BMSCs through the *p38 MAPK* pathway. Clinically, cell exposure to cold can be classified into four grades: mild (32–35 °C), moderate (28–32 °C), severe (20–28 °C), and deep (20–1 °C) [63]. Liu [64] found that that BMSCs exposed to cold conditions (33 °C, 18 °C) for 24 h showed low proliferative activity and almost no differentiation, but were more tolerant to hypoxic environments. The above data suggest that exposure to the cold environment causes spindle-like and fibroblast-like changes in BMSCs and reduces the proliferation of BMSCs, but the cold environment also promotes osteogenic differentiation of BMSCs.

OBs are derived from BMSCs, which are responsible for new bone formation. Sequential expression of several transcription factors is required to activate the OB differentiation pathway during skeletal development. Among these is *Runx2*, which is expressed at several early stages of OB differentiation, and is essential for OB differentiation and chondrocyte maturation. It is also required for the proliferation of preosteoblasts and MSCs throughout the skeleton by increasing the proliferation of undifferentiated MSCs or osteoblastic progenitors, which form a cohesive cellular layer and promote their differentiation to osteoblastic lineage cells [65]. Motyl [47] showed that *Runx2* was significantly inhibited in cold (4 °C) mice. Consistent with this, Aisha [66] found that exposure of normal human OBs (NHOst) to acute hypothermia (27 °C) significantly downregulated *Runx2* expression, suggesting that cold may inhibit the proliferation of OBs. In addition, their cell experiments also showed that cryotreated NHOst had a variable cytoskeleton, and the severity of the changes depended on the severity and duration of the low temperature. Not only that, temperature also affects NHOst mitochondrial activity. NHOst cells exposed to mild hypothermia (35 °C) were able to maintain the integrity of the cytoskeleton by increasing actin and microtubulin fibres but also had increased mitochondrial activity compared to NHOst cells grown under physiological conditions (37 °C). However, NHOst mitochondrial activity was inhibited after 1 h of exposure to severe hypothermia (27 °C) and remained reduced for 72 h, suggesting that the cells may be in a quiescent or senescent stage, or actually dead. This may be related to the reduction in energy expenditure caused by the usual inhibition of protein synthesis at severely low temperatures. All of the above data suggest that low temperature exposure leads to perturbation of the NHOst cytoskeleton and affects the mitochondrial activity of NHOst. NHOst cellular metabolism is affected by the severity of low temperatures, which may be related to the fact that the cold shock protein RNA-binding motif protein 3 (RBM3) is increased during hypothermia [66]. Importantly, overexpression of RBM3 increased the expression of osteogenic genes such as *Runx2* and OCN, as well as the phosphorylation of extracellular signal-regulated kinase (ERK) and *p38*
*MAPK*, promoting OB production [67]. Although overexpression of RBM3 promotes OB formation in cold environments, as a whole, OB production still shows a downward trend in overall OB production [66]. In addition, it has been demonstrated that cold stimulation also affects OBs by promoting the production of BAT, which upregulates the levels of ALP, OCN, and *Runx2* and promotes osteogenic differentiation, as shown above. Meanwhile, Du [46] also found that BAT can secrete cytokines, such as leptin, a hormone with direct anabolic effects on OB [68]. All these confirm that cold promotes the proliferation of OBs through BAT. From the above experiments, we can find that the regulation of OBs by cold stimulation is multifaceted and complex, and the final result depends on the combined effects of various aspects, but in general, the cold environment will reduce the proliferation of OBs.

Osteocytes are highly differentiated cells derived from OBs, accounting for more than 90% of the total cells in the adult skeleton and playing a crucial role in bone tissue; they are mechanosensory cells that control bone remodelling and are pushed towards bone deposition or resorption based on mechanical cues [69]. Du [70] found that exposure to cold (4 °C) led to apoptosis of OBs, and that dead OBs exhibited higher RANKL expression in neighbouring cells. Meanwhile, Suto [71] found that in the frozen state (−80 °C), many atrophied OBs contained condensed chromatin granules that were densely arranged at the inner edge of the nuclear membrane, similar to OBs, but more sparsely distributed and usually located in the centre of the nucleus. In addition, subcellular organelles in many OBs were disrupted by ice crystals. Surprisingly, a small proportion of OBs maintained relatively large nuclei with double membranes and intact cell membranes and cytoplasm. Du [70] also demonstrated that E11/podoplanin was reduced within 14 days in the cold-treated group. E11/podoplanin, a cell membrane protein that can be expressed in OBs, is essential for osteoblast differentiation [72] and dendrite formation in osteoblasts, and cold decreases E11/podoplanin, resulting in altered dendritic differentiation of osteoblasts in the skeletal system of rats [73]. Furthermore, in the skeletal system, rat ECs and vascular-associated reticulocytes play an important role in bone development and fracture healing processes [74]. Additionally, VEGF, also known as vascular permeability factor, plays a key role in promoting the spread of ECs due to its effect on increasing vascular permeability [75]. Zhang [76] found that exposure to a cold environment (−5 °C) increased microvascular permeability and VEGF levels, leading to plasma leakage of large proteins and endothelial damage. ECs regulate regional blood flow [77]. In patients with osteonecrosis, a decrease in ECs due to the cold environment leads to a decrease in blood supply and death of OCs [78]. In addition, ECs have been shown to be effective in fracture healing and repair. In addition, an active blood supply is essential for healing tissue formation during fracture healing and repair [79]. Blood flow plays an important role in angiogenesis [80]. Reduced blood flow leads to impaired angiogenesis in fractures and delays bone healing and regeneration [81]. Reduced blood flow leads to impaired vascularisation in fractures, delaying healing and regeneration. Therefore, cold reduces the proliferation of osteoblasts.

OCs are multinucleated cells of the monocyte–macrophage lineage responsible for remodelling the skeleton by resorbing bone [82], working in coordination with OBs in bone remodelling. Their differentiation involves several stages: OC precursors (OCPs), mononuclear OCs, multinuclear OCs, and mature OCs [83]. Du [46] found that the number of OCs/bone surface increased in mice at day 14 in the cold (4 °C). Meanwhile, Motyl [47] also found that OC surface, OC number, and the percentage of eroded surface were significantly higher in cold (4 °C) mice, and the expression of the OCs recruitment factor RANKL was elevated by cold exposure [84]. Consistent with this, Singh [85] found that cold stress (8 °C and 4 °C) promoted free radical production in fresh bone marrow cells, which enhanced RANKL expression and contributed to OC formation. In addition, cold stimulation affects OCs by promoting the production of BAT, which secretes the cytokine IL-6 [86]. IL-6 indirectly promotes the formation of OCs and bone resorption by promoting RANKL expression. The above studies suggest that cold stimulates the proliferation of OCs (Figure 4).

### 5.2. Effects of Cold Environments on Bone Vascularity

The vasculature is a multifunctional transport system that plays a crucial role in organ development, regeneration, and stem cell behaviour [87]. Bone is highly vascularised and receives about 10–15% of resting cardiac output [88]. Ramasamy [74] classified the intraosseous microvasculature into two subtypes based on the morphology, surface markers, and functional characteristics of the ECs: H-type and L-type. The H-type vessels are specialised vessels found in the bone marrow that are closely related to osteogenic activity. They are characterised by high expression of endostatin and CD31 [89]. In addition to VEGF mentioned above, PDGF-BB, SLIT3, *hypoxia-inducible factor 1-alpha (HIF1-α)*, *Notch*, and other factors are also critical for H-type angiogenesis [90]. Park [91] found that NPY deficiency resulted in impaired survival of HSCs. This suggests that cold exposure may inhibit NPY and thus reduce the number of HSCs. In addition, HSCs in the bone marrow can differentiate into OC precursor cells, which can secrete PDGF-BB that enhances H-vessels during bone modelling and remodelling [90,92,93]. Therefore, the reduction of HSCs due to cold exposure may have an effect on H-vessel production by decreasing the secretion of OC precursor cells and, to some extent, by decreasing the secretion of PDGF-BB, which ultimately inhibits bone growth. In addition, the cold environment also affects H-type angiogenesis by inhibiting the proliferation of OBs, leading to a decrease in OB-derived SLIT3, which further inhibits bone formation. SLIT3 promotes H-type angiogenesis and bone formation [90]. In addition, cold-induced reduction in blood flow decreases *Notch* signalling, leading to reduced bone mass and defective angiogenesis [94]. In summary, the cold environment interferes with angiogenesis and inhibits bone growth by affecting factors such as PDGF-BB, SLIT3, *Notch*, and VEGF (Figure 5).

### 5.3. Effect of Cold Environment on Bone Fat

OBs and adipocytes in bone marrow originate from BMSCs. The balance between adipocyte and OB differentiation in this cell population changes with age, environment, and clinical diseases such as osteoporosis [95]. In this population, an increased production of bone fat relative to OBs leads to decreased bone mass and quantity [96]. MAT is involved in systemic metabolism by producing hormones such as lipocalin and leptin, mainly localising in trabecular compartments [97,98]. There are two main types of MAT: constitutive MAT (cMAT) and regulatory MAT (rMAT). cMAT is a dense accumulation of adipocytes in the early stages of MAT maturation, which are relatively devoid of active haematopoiesis. In contrast, rMAT is MAT that continues to accumulate in the red haematopoietic bone marrow region after the initial peak of early MAT formation [99]. Scheller [98] found that rMAT levels in the growth plate and tibial epiphysis at the tibia/fibula junction were reduced by 76% in 12-week-old mice and by 71% in the proximal tibia after cold exposure (4 °C). In 56-week-old mice, rMAT decreased by 56% and 71%, respectively. In contrast, cMAT below the fibular attachment in the distal tibia remained unchanged. MAT loss in the proximal tibial metaphysis was most prominent in the centre of the bone marrow gap, whereas adipocytes directly adjacent to the inner cortical surface remained unchanged. This is in contrast to the developmental pattern of proximal MAT accumulation. Tibial trabeculae and cortical parameters remained essentially unchanged despite substantial rMAT loss. In conclusion, cold decreases bone marrow adipocytes. At the same time, during cold exposure, the sympathetic nervous system is activated, and NE is released into the circulation and BAT, where it activates *cAMP* [100]. *CAMP* plays a major role in lipolysis and determines the rate of lipolysis in adipocytes and adipose tissue [101]. Activation of the *cAMP-PKA* signalling pathway mobilises bone marrow fat via lipolysis to drive differentiation and function of mesenchymal and HSCs [102]. Hypothermia greatly increases plasma glycerol concentrations in humans and *cAMP* concentrations in rats [103]. Consistent with this, Nie [52] also found that *cAMP* levels increased with the duration of cold exposure. All of the above data suggest that prolonged cold exposure can increase the rate of lipolysis in adipose tissue to some extent. In addition, *proliferator-activated receptor-γ (PPAR-γ)* is often referred to as a major regulator of adipogenesis [104]. *Proliferator-activated receptor-γ coactivator (PGC)-1α* can assist in activating the gene expression of a range of nuclear receptors, including *PPAR-γ* [105]. Cold exposure during exercise increases *PGC-1α* in rats, and cold stimulation alone cannot produce this increase. Thus, cold exposure combined with physical activity appears to be necessary to cause a strong induction of *PGC-1α* expression [106]. It is suggested that the cold environment, combined with exercise, may activate gene expression of *PPAR-γ* by inducing *PGC-1α*, ultimately contributing to the differentiation of BMSCs towards adipocytes.

Based on the above studies, we found that exposure to cold causes spindle-like and fibroblast-like changes in BMSCs and decreases their proliferation. Cold exposure also promotes osteogenic differentiation of BMSCs and inhibits the expression of *Runx2*, which reduces the proliferation of OBs and enhances the expression of RANKL, which contributes to the formation of OCs. In addition, cold reduces the transmembrane glycoprotein E11/podoplanin, which alters the dendritic differentiation of OBs, and increases the level of vascular VEGF, which causes damage to ECs, leading to a decrease in blood supply and, consequently, a decrease in the number of OBs. It also affects vascular function by influencing factors such as PDGF-BB, SLIT3, *Notch*, VEGF. Furthermore, cold promotes lipolysis of MAT through *cAMP* and combines with motility to cause the differentiation of BMSCs towards adipocytes (Figure 6).

### 5.4. Effects of the Cold Environment on the Bony Nerves and Lymphatic

The literature related to the effects of cold environments on osteoneural and lymphatic systems has not been reviewed.

Under normal temperatures, BMSCs display high proliferative capacity and balanced differentiation into OBs, adipocytes, or chondrocytes [107], primarily via *Wnt/β-catenin* and *BMP* signalling pathways [108]. *Runx2* and *Osterix* upregulate Col1 and OCN secretion, while OC activity is modulated by the OPG/RANKL ratio to maintain bone homeostasis [109]. Osteocytes mechanosense via dendritic networks to regulate bone turnover [110], H-type vessels support osteoprogenitor recruitment [74], and MAT secretes adiponectin to preserve bone mass [97]. Cold stress suppresses these factors, impairing microenvironmental function.

There are limitations to the current studies. There is no consensus on temperature thresholds, duration, and frequency of cold exposure for bone health. Future studies should establish standardized protocols to distinguish between positive and negative effects. For example, short-term cold exposure may improve bone metabolism by activating BAT [46], while long-term exposure may exacerbate bone loss [5]. Dose-response relationships need to be further explored in the future to provide evidence for clinical and occupational protection. In addition, the synergistic effects of exercise intervention also need further investigation. Exercise may alleviate bone loss caused by cold exposure by promoting growth plate solute transport [20] and activating *PGC-1α* to enhance osteogenesis [106]. However, the interaction mechanism of exercise intensity, cold adaptation, and bone remodelling is still unclear. It is necessary to explore the comprehensive effects of the cold environment and other factors (such as nutrition, exercise, etc.) on bones in order to fill these gaps in knowledge and provide a more solid theoretical basis for the prevention and treatment of cold-related bone diseases.

## 6. Conclusions

The cold environment has multiple effects on bone. It can reduce bone length, bone thickness, and BMD. The role of BAT is very complicated; it can promote bone resorption by secreting cytokines such as leptin, FGF21, IL-6, etc, but it can also promote bone formation and inhibit bone resorption by increasing the expression level of UCP1 and upregulating the levels of ALP, OCN, and *Runx2*. Most experiments suggest that BAT has a positive effect on the regulation of bone metabolism, promoting bone formation and inhibiting bone resorption. Secondly, cold can increase the elasticity and mechanical support of the constituent materials of bone, decrease bone strength, and ultimately affect the biomechanical properties of bone. Furthermore, cold environments may promote bone resorption by activating sympathetic nerves and regulating leptin secretion, and regulate bone metabolism by activating the *P38 MAPK* signalling pathway and increasing BAT synthesis. Finally, the effects of the cold environment on the bone microenvironment include seven main aspects. First, cold stimulation alters BMSCs and decreases their proliferation, while promoting their osteogenic differentiation. Second, the cold environment also reduces the proliferation of OBs, while promoting the formation of OCs and leading to a decrease in OBs. In addition, cold exposure impairs H-vessels and ultimately inhibits bone growth. Cold also activates *cAMP* to promote MAT catabolism. In addition to MAT, BAT in adipose tissue is also involved in the regulation of bone metabolism in the cold.

Based on the literature review of the effects of the cold environment on bone, it is found that there is a lack of clear criteria regarding the specific temperature, time, and application site of cold stimulation. Therefore, it is important to refine these criteria to harness the effective effects of exercise on bones in cold environments, study the characteristics of bones from various aspects, avoid the adverse effects of the cold environment on bones, and make full use of the positive effects of cold environments in the treatment of bone-related diseases (Table 1).

## Figures and Tables

**Figure 1 biomolecules-15-00564-f001:**
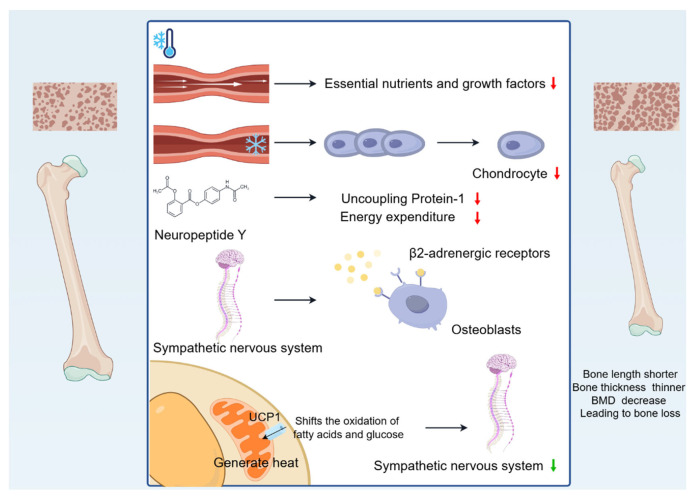
Changes in bone mass caused by cold environment. Cold-induced vasoconstriction causes a reduction in blood flow, leading to a reduced supply of essential nutrients and growth factors, and the reduced blood supply limits bone growth. Vasoconstriction decreases the temperature of developing cartilage, reducing chondrocyte proliferation, destroying the matrix, and slowing the rate of skeletal growth. Cold exposure affects the production and release of neurotransmitters, such as NPY, which inhibits energy expenditure. In addition, cold activates the sympathetic nervous system, increases sympathetic tone, and leads to bone loss through hypothalamic signalling of β2-adrenergic receptors on OBs. Interestingly, UCP1 activation in BAT shifts the oxidation of fatty acids and glucose to generate heat, reducing sympathetic tone and alleviating bone loss.

**Figure 2 biomolecules-15-00564-f002:**
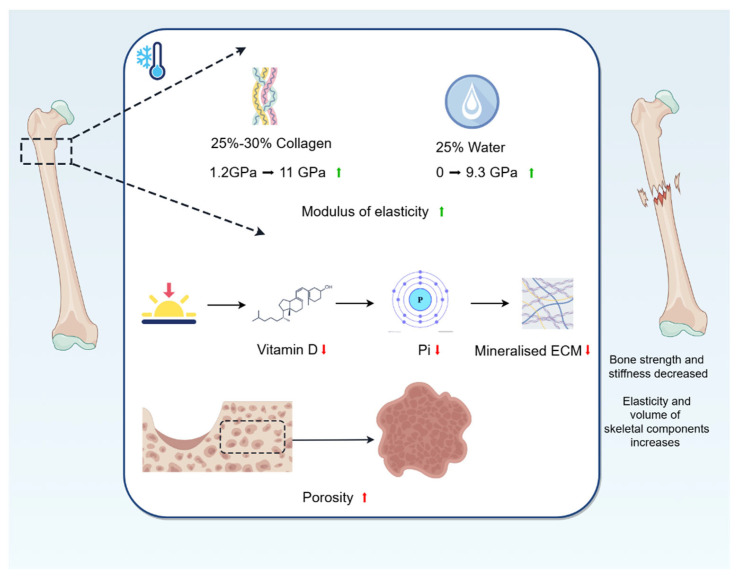
Changes in bone biomechanics caused by cold environment. The modulus of elasticity of collagen fibres, which make up 25–30%, rises from 1.2 GPa (25 °C) to 11 GPa (−18 °C); and the modulus of elasticity of water, which makes up 25%, rises from 0 GPa (23 °C) to 9.3 GPa (−16 °C). A cold environment improves the elasticity and increases the volume of some of the bone constituents, providing mechanical support to the surrounding tissues. Being in a cold region may lead to vitamin D deficiency, which reduces the uptake of Pi, and this reduction in Pi affects mineralisation of ECM, leading to bone softening and ultimately affecting the biomechanical properties of bones. In addition, cold causes bone loss, decreases BMD, and thus increases porosity, which ultimately affects the elastic modulus and strength of bone.

**Figure 3 biomolecules-15-00564-f003:**
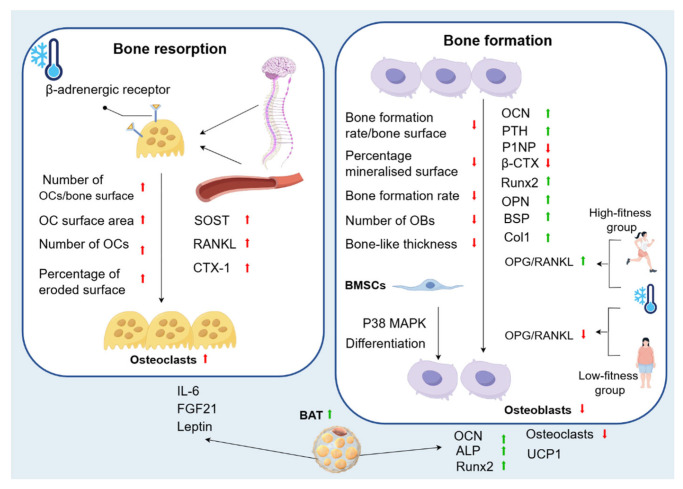
Changes in bone metabolism caused by cold environment. Cold exposure promotes bone loss by inhibiting osteogenesis and enhancing osteoclastic metabolism. In bone resorption, there is an increase in the number of OCs/bone surface, in addition to a significant increase in OC surface area, number of OCs, and percentage of eroded surface. Cold can also affect biomarkers of bone remodelling and blood-bone immune crosstalk, with an increase in SOST, RANKL, and CTX-1. This is because sympathetic nerve activation by low temperature leads to β-adrenergic receptor activation and subsequent initiation of OCs-associated bone resorption. In bone formation, the bone formation rate/bone surface decreases significantly, as well as a significant decrease in the percentage of mineralised surface, bone formation rate, number of OBs, and bone-like thickness. In addition, OCN levels are increased. With WBC, OPG/RANKL increases in the high-fitness group and decreases in the low-fitness group. PTH increases significantly after cold, while P1NP and β-CTX decreases significantly. Cold exposure also significantly upregulates *Runx2* in the process of osteogenic differentiation of MSC, as well as the expression of OPN, BSP, and Col1. At the same time, phosphorylation of the *p38 MAPK* pathway is activated to promote osteogenic differentiation of BMSCs. This complex phenomenon is explained by the fact that cold exposure induces changes in bone metabolism and morphology by increasing sympathetic tone and increasing BAT. BAT can accelerate bone resorption through the secretion of FGF21, IL-6, and leptin. On the other hand, the regulation of bone metabolism by BAT is positive in humans, as it upregulates the levels of ALP, OCN, and *Runx2*, promotes osteogenic differentiation, and can reduce the resorption capacity of OCs, diminishing their resorption capacity.

**Figure 4 biomolecules-15-00564-f004:**
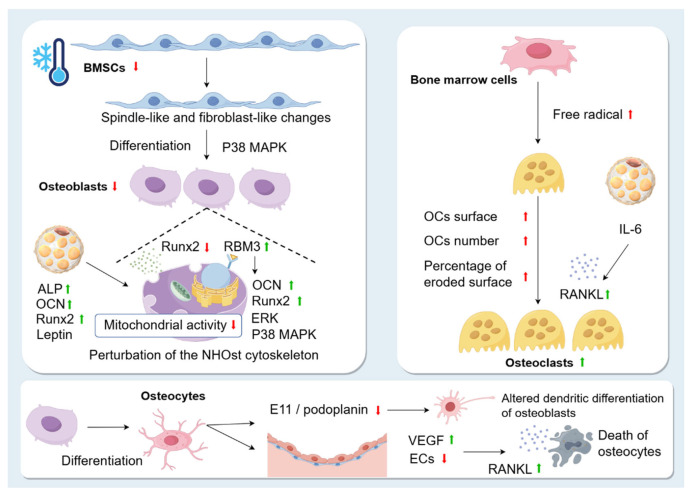
Changes in skeletal cells caused by cold environment. Exposure to the cold environment causes spindle-like and fibroblast-like changes in BMSCs and reduces their proliferation. However, the cold environment also promotes osteogenic differentiation of BMSCs through the *p38 MAPK* pathway. *Runx2* is significantly inhibited in the cold. Low temperature exposure leads to perturbation of the NHOst cytoskeleton and affects the mitochondrial activity of NHOst. This may be related to the fact that the cold shock protein RBM3 is increased during hypothermia. Overexpression of RBM3 increases the expression of osteogenic genes, such as *Runx2* and OCN, and promotes the phosphorylation of extracellular signal-regulated kinase (ERK) and *p38*, promoting OB production. In addition, cold stimulation affects OBs by promoting the production of BAT, which upregulates the levels of ALP, OCN, leptin, and *Runx2* and promotes osteogenic differentiation. Osteocytes are highly differentiated cells derived from OBs. Exposure to cold leads to apoptosis of OBs, and dead OBs exhibit higher RANKL expression in neighbouring cells. Cold decreases E11/podoplanin, resulting in altered dendritic differentiation of osteoblasts in the skeletal system of rats. Exposure to a cold environment increases microvascular permeability and VEGF levels, leading to plasma leakage of large proteins and endothelial damage. In patients with osteonecrosis, a decrease in ECs due to the cold environment leads to a decrease in blood supply and death of OCs. OC surface, OC number, and percentage of eroded surface are significantly increased in the cold environment. Cold stress promotes free radical production in fresh bone marrow cells, which enhances RANKL expression and contributes to OC formation. In addition, BAT secretes the cytokine IL-6, which indirectly promotes the formation of OCs and bone resorption by promoting RANKL expression.

**Figure 5 biomolecules-15-00564-f005:**
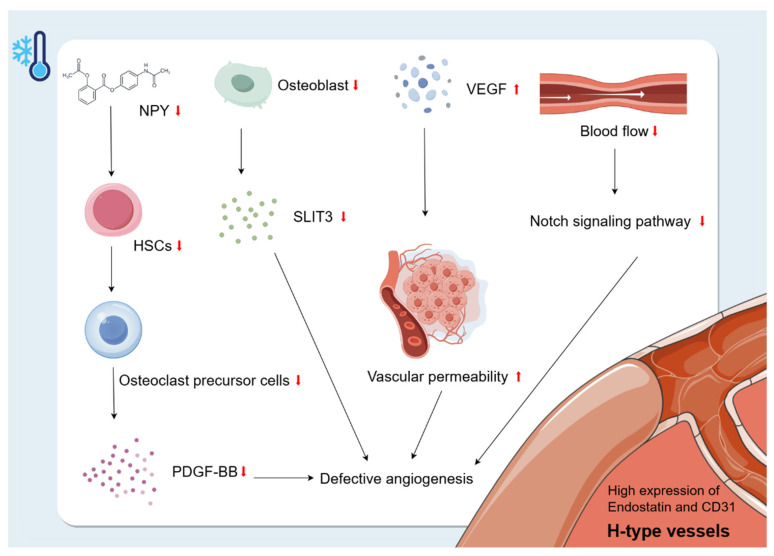
Changes in bone vascularity caused by cold environment. The H-type vessels are specialised vessels found in the bone marrow that are closely related to osteogenic activity. They are characterised by high expression of endostatin and CD31. Exposure to a cold environment increases microvascular permeability and VEGF levels, leading to plasma leakage of large proteins and endothelial damage. Cold exposure may inhibit NPY and thus reduce the number of HSCs. The reduction of HSCs due to cold exposure may have an effect on H-vessel production by decreasing the secretion of OC precursor cells and, to some extent, decreasing the secretion of PDGF-BB, which ultimately inhibits bone growth. In addition, the cold environment also affects H-type angiogenesis by inhibiting the proliferation of OBs, leading to a decrease in OB-derived SLIT3, which further inhibits bone formation. In addition, cold-induced reduction in blood flow decreases *Notch* signalling, leading to reduced bone mass and defective angiogenesis.

**Figure 6 biomolecules-15-00564-f006:**
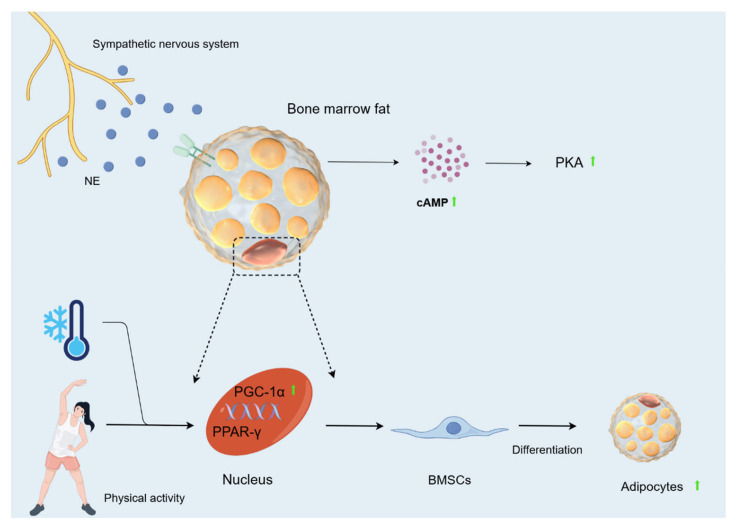
Changes in bone fat caused by cold environment. During cold exposure, the sympathetic nervous system is activated, and NE is released into the circulation and BAT, where it activates *cAMP*. Activation of the *cAMP-PKA* signalling pathway mobilises bone marrow fat via lipolysis to drive the differentiation and function of mesenchymal cells. Hypothermia greatly increases plasma glycerol concentrations in humans and *cAMP* concentrations in rats. The cold environment, when combined with exercise, may activate gene expression of *PPAR-γ* by inducing *PGC-1α*, ultimately contributing to the differentiation of BMSCs towards adipocytes.

**Table 1 biomolecules-15-00564-t001:** Comparative bone characteristics under normal versus cold temperature conditions.

Feature	Normal Temperature	Cold Environment
Bone mass	Normal bone length and thickness, stable BMD.	Shortened bone length, thinner bone thickness, decreased BMD, resulting in decreased bone mass.
Bone biomechanics	The biomechanical properties of bone are stable, the mineral content and collagen fibres are orderly arranged, and it can effectively resist external forces and provide good mechanical support.	Bone strength and stiffness decrease, elasticity and volume of some bone components increase.
Bone metabolism	Bone metabolic balance is maintained mainly by the coordinated activity of OBs and OCs. Bone formation markers such as OCN and Col1 have high expression levels, while bone resorption markers such as CTX-1 are low.	Bone metabolic balance is disrupted, and bone resorption activity is enhanced. BAT plays a complex role in bone metabolism regulation. It not only secretes cytokines that promote bone resorption but also promotes bone formation and inhibits bone resorption by increasing certain indicators.
Skeletal cells	MSCs maintain high proliferation rate and differentiate into OBs, adipocytes, or chondrocytes in balance. Osteogenic differentiation mainly depends on *Wnt/β-catenin* and *BMP* signalling pathways. *Runx2* and Osterix are highly expressed, which promotes the secretion of Col1 and OCN. The activity of OCs is regulated by the OPG/RANKL ratio, which cooperates with OBs to maintain bone homeostasis. Osteocytes sense mechanical stimulation through dendritic networks and regulate bone formation/resorption.	BMSCs undergo spindle and fibroblast-like changes, proliferation decreases, but osteogenic differentiation is promoted through the *p38 MAPK* pathway; OBs proliferation decreases; OC proliferation increases; osteocyte apoptosis.
Bone vascularity	H-type blood vessels support the recruitment of osteoblast precursors and provide the necessary conditions for bone growth.	Interferes with angiogenesis and inhibits bone growth.
Bone marrow fat	MAT acts as an energy reserve and secretes adiponectin to protect bone mass.	MAT decreases, promoting the differentiation of BMSCs into adipocytes.

## Data Availability

All data generated or analysed during this study are included in this published article.

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
