# Peer review of "Bone Modelling and Remodelling in Cold Environment"

_biomolecules, 2025, doi:10.3390/biom15040564_

Round 1
Reviewer 1 Report
Comments and Suggestions for Authors
I was very pleased to review the article “Bone Modelling and Remodelling in Cold Environment” This paper certainly helps to improve the research and disease diagnosis related to bone remodeling. I would like to suggest some major revisions to enhance the quality of this paper as follows:
1. Authors explain how a cold environment affects bone remodeling and metabolism. It would be great if they compared their explanation with bone remodeling and metabolism at normal temperatures. Redears can easily compare both situations.
2. Also all figures need to show what happens in normal temperatures and what the different things occur in cold temperatures.
Reviewer 2 Report
Comments and Suggestions for Authors
The topic of the review is interesting because the influence of cold environment on bone modelling and remodelling is worth to be investigated.
The major criticisms is the lack of a proper "future perspective and outlook". The current version of the summary is not providing any critical suggestion as a review should provide.
Round 2
Reviewer 1 Report
Comments and Suggestions for Authors
I was very pleased to review the article “Bone modelling and remodelling in cold environment”. Authors addressed all the points in the revised manuscript I indicated during the review process.
Reviewer 2 Report
Comments and Suggestions for Authors
the authors have addressed the comments